# Kinesiology Taping as an Innovative Measure against Post-Operative Complications after Third Molar Extraction—Systematic Review

**DOI:** 10.3390/jcm9123988

**Published:** 2020-12-09

**Authors:** Aleksandra Jaroń, Maciej Jedliński, Elżbieta Grzywacz, Marta Mazur, Grzegorz Trybek

**Affiliations:** 1Department of Oral Surgery, Pomeranian Medical University in Szczecin, 70-111 Szczecin, Poland; aleksandra.jaron@pum.edu.pl (A.J.); 50422@student.pum.edu.pl (E.G.); grzegorz.trybek@pum.edu.pl (G.T.); 2Department of Interdisciplinary Dentistry, Pomeranian Medical University in Szczecin, 70-111 Szczecin, Poland; 3Department of Oral and Maxillofacial Surgery, Sapienza University of Rome, 00185 Rome, Italy; marta.mazur@uniroma1.it

**Keywords:** kinesiotaping, kinesiology taping, third molar extraction, rehabilitation

## Abstract

There are many randomized clinical trials suggesting a positive effect of kinesiotaping on postoperative swelling. In dentistry, however, the use of kinesiotaping still seems to be innovative, since not many articles on kinesiotaping within the craniofacial area have been published. This study aimed to systematically review and synthesize available controlled trials examining the use of kinesiotaping to reduce morbidity after third molar extraction. Literature searches for free text and MeSH terms were performed using five search engines, and used to find studies which focused on kinesiotaping as a form of rehabilitation after third molar extraction. The keywords used in the search were: “((“molar, third”[MeSH Terms] OR (“molar”[All Fields] AND “third”[All Fields]) OR “third molar”[All Fields] OR (“third”[All Fields] AND “molar”[All Fields])) AND extraction [All Fields]) AND “kinesiology”[All Fields]”. For the assessment of the risk of bias, the Jadad and Maastricht scales were applied. The search strategy identified 317 potential articles. After analysis, 10 papers were included in the final evaluation. Despite the fact that most of the included articles adhered to methodological standards, the fact that there are only a few of them points to a further need for scientific development of physiotherapy in this regard. Kinesiology taping is useful against post-operative morbidity of the third molar extraction site. The present studies show a low level of the risk of bias, but they are limited in number; therefore, it seems that more research is needed.

## 1. Introduction

Patients experience significant disturbances in their everyday life in the five days after third molar surgery [1]. Because the third molars are often unfavorably positioned, and therefore specific procedures must be taken during extraction, complications are often inevitable, being the result of the processes of healing. These unpleasant temporary complications include, first and foremost, swelling, paresthesia, trismus, and pain [2]. This significantly reduces the quality of life of patients [1]. Due to the inconvenience from which patients have to suffer, various methods of alleviating the effects are offered, such as pharmacotherapy [3], local compresses for postoperative sites [4], laser applications [5], or recently, kinesiology taping. Traditionally, practitioners are trying to increase patients’ well-being with the use of pharmacotherapy, such as steroid application pharmacotherapy (e.g., steroid application and painkiller prescriptions). It is no secret, however, that although these drugs are effective, they burden local tissue metabolism [6,7].

Kinesiology taping is one of numerous non-invasive interventions used for posture correction, and is widely recognized in the treatment of musculoskeletal disorders for athletes and the general population [8]. The use of taping as a treatment method was first reported in the literature in 1969 as a helpful measure for rehabilitation of the elbow joint [9]. Since then, taping has gained recognition as a useful tool in the treatment of acute and chronic musculoskeletal complaints, including pain, paresthesia, joint instability, and oedema in different parts of the muscoskeletal system [10]. The tape is made from material which has an elasticity of approximately 130–140%, and is applied to the skin using a certain amount of traction, thereby influencing the skin and various subcutaneous layers. The pre-tension of the tape subtly lifts the skin, thereby possibly improving lymphatic flow and directing it to pathways that suffer less congestion [11,12]. The tape also provides a massaging effect during active movement [12]. The main outcomes of this procedure are normalization of muscle tension, and improvements of blood circulation of small vessels and lymph flow [13]. In sports, to reduce oedema, a number of classic physiotherapeutic interventions are induced, including manual lymphatic drainage and compression treatment using complex multi-layer bandaging or compression stockings, as well as skin care and decongestive exercise [14]. At the 2012 Olympic games in London, taping was one of the five most frequently used treatment modalities, accounting for 8.9% of officially registered interventions, often used for injury prevention [14]. It is additionally believed that in sports medicine, kinesiotaping is as effective in reducing swelling as classic physiotherapeutic methods, while maintaining the compact size of the tape [15], suggesting a positive effect of kinesiotape application on postoperative swelling in a variety of clinical cases [16]. In dentistry, however, the use of kinesiotaping still seems to be innovative, since not many articles on kinesiotaping within the craniofacial area have been published to date, and there is no systematic review of the literature in this field at the moment of writing this paper. This study aimed to systematically review and synthesize controlled trials investigating the use of kinesiotaping to reduce the morbidity after third molar extraction. The results of this review are meant to provide useful information to make clinical decisions, as well as to direct further research in this field.

## 2. Materials and Methods

### 2.1. Search Strategy 

This systematic review was performed according to the PRISMA statement [17] and following the guidelines from the Cochrane Handbook for Systematic Reviews of Interventions [18]. Literature searches for free text and MeSH terms were performed using several search engines: MedLine (PubMed), Scopus, Web of Science, Embase, and Google Scholar search engines were used to find the studies, which focus on application of kinesiotaping as a rehabilitation method after third molar extraction (3 November 2020). All searching was performed using a combination of subject headings and free-text terms—the final search strategy was determined by several pre-searches. The keywords used in the search strategy were as follows: “((“molar, third”[MeSH Terms] OR (“molar”[All Fields] AND “third”[All Fields]) OR “third molar”[All Fields] OR (“third”[All Fields] AND “molar”[All Fields])) AND extraction [All Fields]) AND “kinesiology”[All Fields]”. The cited articles should have explored the subject of rehabilitation method for morbidity, that is, swelling, paresthesia, or pain due to third molar extraction using extraoral kinesiology taping.

### 2.2. Eligibility Criteria

The following inclusion criteria were employed for this systematic review: (1) Randomized clinical trial (RCT); (2) cohort study; (3) case-control study; (4) articles published in the last 10 years (5) published in English; and (6) in-vivo studies. 

The following exclusion criteria were applied: (1) Case reports; (2) reviews; (3) abstracts and author debates or editorials; (4) lack of effective statistical analysis; (5) papers not related to practical implementations of kinesiotaping in rehabilitation of the third molar post-extraction site; and (6) in-vitro studies.

### 2.3. Data Extraction

Titles and abstracts were independently selected by two authors (M.J. and A.J.) following the inclusion criteria. The full text of each identified article was then analyzed to verify whether it was suitable for inclusion. Disagreements were resolved through discussion with the team supervisor (G.T.). Authorship, year of publication, type of each eligible study, and its relevance regarding the use of Kinesio tapes in everyday practice were independently extracted by two authors (A.J. and M.J.) and examined by the third author (G.T.).

### 2.4. Quality Assessment

According to the PRISMA statements, the evaluation of methodological quality gives an indication of the strength of evidence provided by the study because methodological flaws can result in biases [17].

The quality assessment of RCT and RCCT studies was performed using the Jadad scale [19]. In the assessment, it was taken into account whether the study was randomized and double-blinded with appropriately described methods to find the possible risk of bias. A point was given for every characteristic evaluated. The possible assessment was from zero to five, with a high score indicating a good quality of the study. In order to avoid possible homogeneity of the results, as well as the post-rationalization of research methods, possibly adjusted to most popular scales in order to obtain high scores, an additional scale was used. What is more, the Weights Maastricht Scale was also designed especially for physical studies [20]. It consists of 16 components that accurately assess the bias risk, and due to their complexity, they provide a detailed analysis of the results. The maximum number of points that may be obtained is 100, where 100 stands for a methodologically perfect research. The maximum number of points awarded to the research for a given characteristic is included in the table. The methodological score of a study could then be expressed as a range, for example 60–97. This means 60 points for the + (sound methodology), and 3 points subtracted for − (likely sources of bias). In the above example, the study has a range for incomplete or lacking information of 37 (97–60), for 0. This range may be seen as a measure of uncertainty about the quality of the study.

## 3. Results

### 3.1. Study Selection

The search strategy identified 317 potential articles: 26 from PubMed, 2 from Scopus, 3 from Web of Science, 0 from Embase, and 287 from Google Scholar. After duplicates had been removed, 237 articles were analyzed. After that, 223 papers were excluded because they did not meet the inclusion criteria. Of the remaining 14 papers, four were excluded because they were not relevant to the subject of the study. The remaining 10 papers were included in the qualitative synthesis (Figure 1 flow diagram). Table 1 summarizes the characteristics of each of the 10 studies included.

### 3.2. Quality Assessment and the Risk of Bias

The results of the quality assessment are presented in Table 2 and Table 3. All studies ensured that the case and control groups were adequately represented in terms of number, age, and gender. The applied procedures were properly described. In the study of Olivio et al. [31], it was reported that both scales correlated [32]. In our study, a higher score (5) on the Jadad scale was also correlated with a high score on the Maastricht scale. However, the Maastricht criteria shed a different light on the research. 

## 4. Discussion

Generally speaking, most of the studies present a high level in terms of methodology. The authors have made an effort to thoroughly present a new, unproven method in dentistry in the form of good-quality evidence. It is consistent that split-mouth studies provide some randomization and blinding. Making comparisons within one subject, rather than between them, enhances the statistical power [33]. Testing the effects of two different topical therapies on the same patient provides the best evidence when comparing treatment effects. However, by measuring the patient’s sensations, such as pain, there is always a risk of wrong estimation by the patient. In such studies, we must trust the patient’s sincerity as regards to feelings and the determination of well-being, according to the VAS scale. Of course, one cannot fully rule out the placebo effect in such cases. Overall patient reliability is confirmed by the Tatti study, where the placebo-taping group was less able to cope with the healing problems than the one in which subjects were properly taped [27]. The measurement of changes in the facial volume is hard to perform and standardize [34]. However, in each study, the measurement procedure was precisely described. The observer, however, when measuring the swelling, would have never been blinded. This is because the tape is a visible, physical object. Therefore, despite the fact that none of the included studies received points in the K subsection in the Maastricht scale, in the Jadad scale, a point for correct blinding should be awarded. No risk of bias was found in the follow-up phase in any of the studies. This should not be surprising, because with such a frequent and standard procedure as the removal of the third molar, dental surgery specialists are aware of the need for the frequency of follow-up appointments and a proper duration of the follow-up. Chiang et al., however, do not describe on which day tapes were removed. Additionally, with such a short follow-up (seven days in the majority of the studies), it is difficult for a patient to drop out, which increases the quality of research. Extraction of third molars is one of the most frequent surgical procedures performed by Oral and Maxillofacial surgeons. Most often, to relieve the patient during the healing process, pharmacotherapy is applied (i.e., prescription of anti-inflammatory agents) [35]. Ristow, in their study, studied the question of whether kinesiotaping may impact on pain reduction [21]. However, kinesiotaping was reported as a method to reduce pain by lowering the pressure on nociceptors [35]. It seemed to be an interesting alternative to drugs that does not burden the patient’s metabolism or make them dizzy, and can bring comparable results [22,28]. Kinesiotaping is cheap and easy to apply [36]. Nevertheless, it has better results if it is applied by trained staff [37]. Drugs provide greater relief during the first days after surgery than kinesiotaping, but tapes, by causing the pressure in internal fluid flow, resolve the pain faster as a result of faster reduction in inflammatory mediators [23,28]. Researchers do not agree on the time period during which the tape should be applied. Genc removed it after 2 days [23], Tatli, [27]. Gözlüklü, [26] de Rocha Heras [25] and Ristow [21] after 5 days, and Yurttutan after 7 days. There are also proposals for new, less visible [25], and more effective [26] ways of applying the tape in the area of oedema. Most of the studies included in the analysis are very recent reports (from 2019 [22,23] and 2020 [24,25,26,27,28,29,30]). With regret, it must be noted that this research had a small group of patients included in the study, and only one of them (Chiang et al.) scored maximum points on a scale of Maastricht for the number of patients included in the study. This highlights the fact that over the last few years, dentists have shown their growing interest in kinesiology taping. In the craniofacial area, the effectiveness of kinesiotaping in reducing swelling and pain was also found in more severe injuries as zygomatic–orbital fractures [38], mandibular fractures [39], and after orthognathic surgery [40,41]. It was also reported as effective in reducing the symptoms of chronical TMD [42,43].

Despite the fact that most of the articles included in the revision are of high quality and adhere to methodological standards, the fact that they are limited in number (i.e., 10) points to a further need for scientific development of physiotherapy in this regard.

## 5. Conclusions

Kinesiology taping is a useful clinical rehabilitation method against post-operative morbidity of the third molar extraction site. It enables significant reduction of swelling and pain, without burdening the patient’s metabolism nor digestive system. Due to the fact that it acts only locally and is as effective as pharmacotherapy, it should be applied more often in everyday procedures by dental surgeons. The present studies show a low level of the risk of bias, but they are limited in number; therefore, it seems that more research is needed. Subsequent research in this area should not only include more patients in every new study, but also try to applicate kinesiotaping over other types of intervention in dental surgery. In this way, the effectiveness of this method will be even more proven, and thus will gain support and popularity among practitioners.

## Figures and Tables

**Figure 1 jcm-09-03988-f001:**
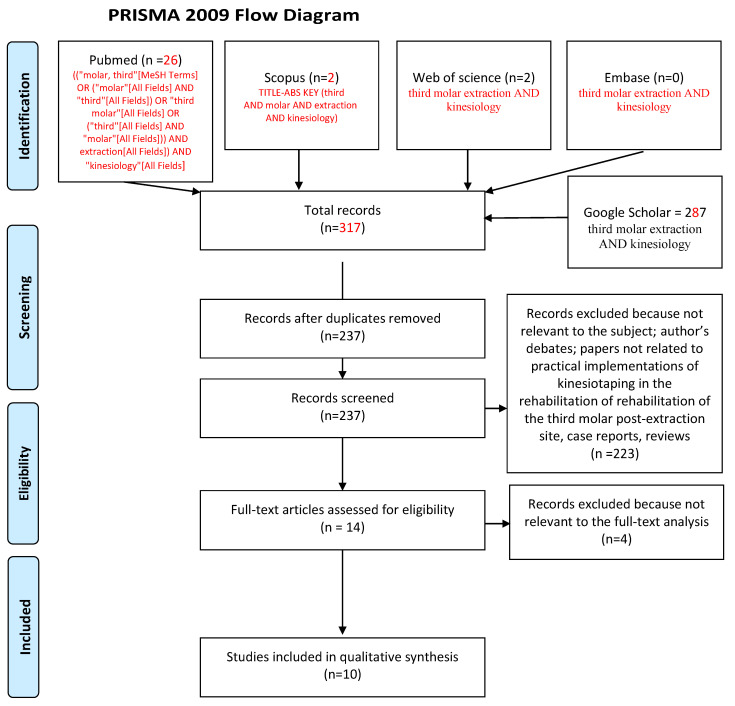
Prisma 2009 flow diagram representing the study selection process.

**Table 1 jcm-09-03988-t001:** Characteristics of included studies.

Author and Year of Publication	Type of Study	Study Objective	Number of Subjects	Test Group	Control Group	Follow Up	Study Characteristics	Results
**Ristow et al. (2014) [21]**	RCT	Investigation of post-operative morbidity reduction in individuals subjected to surgical extraction of the third molar	40 (19 F, 21 M)	Kinesiology Taping	No intervention	7 days	Procedure under GA Tapes remained for 5 days Examined variables = trismus, pain (VAS), swelling	Swelling-decrease Pain-decrease Trismus-decrease
**Mohammed and Delemi (2019) [22]**	RCT	Comparison of kinesiology taping effects to pharmacotherapy (Diclofenac) on reducing post-operative morbidity after third molar surgery	30 (13 F, 17 M)	Kinesiology Taping	Diclofenac assumption	7 days	Procedure under LA Tapes applied for 7 days Examined variables= swelling	Swelling-decrease
**Genc et al. (2019) [23]**	RCT split-mouth study	Comparison of the effects of the surgical drain and kinesiotape applications on postoperative morbidity after mandibular third molar surgery	23 (12 F, 11 M)	Kinesiology Taping	Surgical tube drain	7 days	Procedure under LA Tapes applied for 3 days, drain removed after 2 days Examined variables = swelling, trismius, pain (VAS)	Swelling-increase Trismus-similar Pain-similar
**Erdil et al. (2020) [24]**	RCT	Comparison of the effects of preoperative submucosal corticosteroid injection and postoperative KT application with postoperative non-steroid anti-inflammatory therapy on postoperative inflammatory symptoms	52 (36 F, 16 M)	Kinesiology taping + paracetamol 500 mg	1 group Dexamethasone injection submucosal (8 mg/2 mL) + Paracetamol 500 mg 2 group 25 mg dexketoprofen trometamol prescription. + Paracetamol 500 mg	7 days	Procedure under LA Tapes applied for 3 days Examined variables = swelling, trismius, pain, QoL	Swelling-decrease Trismus-decrease Pain-increase QoL-increase The combination of KT with pharmacotherapy would be more useful after third molar surgeries.
**de Rocha Heras et al. (2020) [25]**	RCT split-mouth study	Investigation of pain and edema reduction in individuals in which extraction of impacted mandibular third molars was performed	13 (8 F, 5 M)	Kinesiology taping	None	5 days	Anesthesia: not reported Tapes applied for 5 days Examined variables = swelling, pain (VAS)	Swelling-decrease Pain-decrease
**Gözlüklü et al. (2020) [26]**	RCT split-mouth study	Comparison of effects of two kinesiology taping techniques after third molar extraction	60 (33 F, 27 M)	The base of 3 strips of equal length was placed above the supraclavicular lymph nodes using original method of taping.	In addition to the classical technique a masseteric support bandage was placed.	7 days	Procedure under LA Tapes applied for 5 days Examined variables = swelling, trismus, pain (VAS)	Swelling-unclear Trismus-decrease Pain-unclear
**Tatli et al. (2020) [27]**	RCT	Investigation of pain and edema reduction in which extraction of impacted mandibular third molars was performed	60 (non reported)	Kinesiology Taping	1st group placebo taping 2nd group none intervention	7 days	Procedure under LA Tapes applied for 5 days Examined variables = swelling, trismius, pain (VAS)	Swelling-decrease Trismus-decrease Pain-decrease
**Mohammed and Delemi (2020) [28]**	RCT	Comparison of kinesiology taping effects to pharmacotherapy (submucosal Dexamethasone injection) on reducing post-operative morbidity after third molar surgery	30 (13 F, 17 M)	Kinesiology Taping	submucosal Dexamethasone injection	7 days	Procedure under LA Tapes applied for 7 days Examined variables = swelling, pain (VAS)	Swelling-similar Pain-similar
**Chiang et al. (2020) [29]**	RCT	Investigation of post-operative morbidity reduction in which extraction of impacted mandibular third molars was performed	76 (46 F, 30 M)	Kinesiology Taping + analgesics + antibiotics	Analgesics +antibiotics	7 days	Anesthesia not reported Tapes applied for 7 days Examined variables = swelling, pain (VAS)	Swelling-decrease Trismus-decrease Pain-decrease QoL-increase
**Yurttutan and Sancak (2020) [30]**	RCT, split mouth	Investigation of post-operative pain and swelling reduction in individuals subjected to surgical extraction of the third molar	60 (gender not reported)	Kinesiology Taping + analgesics + antibiotics	Analgesics + antibiotics	7 days	Procedure under LA Tapes applied for 7 days Examined variables = swelling, trismus, pain (VAS), QoL	Swelling-decrease Pain-decrease

**Table 2 jcm-09-03988-t002:** Quality assessment according to the Jadad scale.

Jadad Scale for Reporting Randomized Controlled Trials [15]					
Author	Ristow et al. [21]	Mohammed and Delemi [21]	Genc et al. [23]	Erdil et al. [24]	de Rocha Heras et al. [25]
Randomization present	1	1	1	1	1
Appropriate randomization used	1	1	1	1	1
Blinding present	0	0	1	1	1
Appropriate blinding used	0	0	1	0 *	1
Appropriate long-term follow-up for all patients	1	1	1	1	1
Total	3	3	5	4	5
Author	**Gözlüklü et al.** [26]	**Tatli et al.** [27]	**Mohammed and Delemi** [28]	**Chiang et al.** [29]	**Yurttutan and Sancak** [30]
Randomization present	1	1	1	1	1
Appropriate randomization used	1	1	1	1	1
Blinding present	1	0	0	0	0
Appropriate blinding used	1	0	0	0	0
Appropriate long-term follow-up for all patients	1	1	1	0	1
Total	5	3	3	2	3

*—insufficiently described.

**Table 3 jcm-09-03988-t003:** Quality assessment according to the Maastricht scale.

	Points									
Weights	Ristow et al. [21]	Mohamm-ed and Delemi [22]	Genc et al. [23]	Erdil et al. [24]	de Rocha Heras et al. [25]	Gözlüklü et al. [26]	Tatli et al. [27]	Mohamm-ed and Delemi [28]	Chiang et al. [29]	Yurttu-tan and Sancak [30]
A. Selection and retrestiction										
1. Description of inclusion and exclusion criteria 2 pts.	2	2	2	2	2	2	2	2	2	2
2. Restriction to a homogeneous study population 2 pts.	2	2	2	2	2	2	2	2	2	2
B. Treatment allocation										
1. Randomization if yes, than	1	1	2	2	2	2	2	2	2	2
2. Allocation procedure not leading to bias 10 pts.	10	10	10	10	10	10	10	10	10	10
3. Blinded allocation procedure 5 pts.	0	5	5	5	5	5	0	0	0	0
C. Study size										
1. Smallest group bigger than 25 subjects 4 pts.	4	4	4	4	-	4	4	4	4	4
2. Smallest group bigger than 50 subjects 4 pts.	-	-	-	4	-	-	4	-	4	4
3. Smallest group bigger than 75 subjects 4 pt.	-	-	-	-	-	-	-	-	4	-
D. Prognostic comparability										
1. −5.9 pts.	9	9	9	9	9	9	9	9	9	9
E. Drop-outs										
1. No dropouts, 12 pts.	12	-	-	12	12	12	12	12	12	12
2. Number of drop-outs given in each group each group 2 pts.	0	0	0	-	-	-	-	-	-	-
3. Reasons for withdrawal (of drop-outs) given in each group 2	0	0	0	-	-	-	-	-	-	-
4. Dropouts not leading to bias (less than 5% drop-outs) 8 pts.	0	0	8	-	-	-	-	-	-	-
F. Loss to follow-up										
1. less than 20% loss to follow-up in all groups 2 pts.										
2. less than 10% loss to follow-up in all groups 2 pts.	8	0	8	8	8	8	8	8	8	8
3. loss to follow-up not leading to bias (less than 5%) 8 pts.										
G. Intervention 6 pts. For proper description	6	6	6	6	6	6	6	6	6	6
H. Extra treatments										
1. No co-interventions, or 2 pts.	2	2	2	2	2	2	2	2	2	2
2. Co-interventions comparable between groups 2 pts.										
I. Blinding of patient										
1. Attempt at blinding or naive patient 2 pts.	-	-	2	2	2	0	-	-	-	-
2. Blinding evaluated and successful 2 pts.	-	-	2	2	2	0	-	-	-	-
J. Blinding of therapist										
1. Attempt at blinding 2 pts.	-	-	2	2	2	2	-	-	-	-
2. Blinding evaluated and successful 2 pts.	-	-	2	2	2	2	-	-	-	-
K. Blinding of observer										
1. Attempt at blinding 2 pts.	-	-	-	-	-	-	-	-	-	-
2. Blinding evaluated and successful 2 pts.	-	-	-	-	-	-	-	-	-	-
L. Outcome measures 5 pts.	5	5	3	5	5	5	5	3	3	5
M. Follow-up period										
1.Measurement just after the last treatment 1pt.	1	1	1	1	1	1	1	1	1	1
2. Timing comparable 1 randomisation (if relevant) 1 pt.	1	1	1	1	1	1	1	1	0	1
3. Measurement three months or longer after randomisation (if relevant) 1 pt.	0	0	0	0	0	0	0	0	0	0
N. Side effects 1 pt.	0	1	0	1	0	0	0	0	0	0
O. Analysis and presentation of data										
1. Frequencies 1 pt.	1	1	1	1	1	1	1	1	1	1
2. Intention to treat, or 3 pts.	3	3	3	3	3	3	3	3	3	3
3. Corrections for baseline differences, non-compliance, and drop-outs 3 pts.	3	2	3	3	2	3	3	2	3	3
P. Total	69–81	51–80	76–81	87–91	76–81	81–86	73–77	69–75	75–81	73–77

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
