# Peer review of "Kinesiology Taping as an Innovative Measure against Post-Operative Complications after Third Molar Extraction—Systematic Review"

_jcm, 2020, doi:10.3390/jcm9123988_

Round 1

Reviewer 1 Report

ABSTRACT

line 14  The abbreviation should be given at the first mention in the manuscript. Randomized clinical trial (RCT)

The date of search does not have to be given in abstract, but methods. Date should be without full stops.

RESULTS

Quality assessment and the risk of bisa are presented in tables. The argumentation and discussion (lines 123-147) about this would maybe better suit into DISCUSSION.

DISCUSSION

Line 167 references are not in brackets

Author Response

Reviewer 1

Dear Reviewer,

We would like to thank you for your valuable comments on the article. Below you will find our reply to your review. All changes are with a description or a comment and changes have been made to the manuscript

ABSTRACT

line 14  The abbreviation should be given at the first mention in the manuscript. Randomized clinical trial (RCT)

Thank you for comment. I introduced your suggestion to the text.

The date of search does not have to be given in abstract, but methods. Date should be without full stops.

Thank you for remark. We introduced it to the text.

RESULTS

Quality assessment and the risk of bias are presented in tables. The argumentation and discussion (lines 123-147) about this would maybe better suit into DISCUSSION.

 Thank you for remark. We introduced it to the text.

DISCUSSION

Line 167 references are not in brackets

Thank you for comment. We corrected this error.

Reviewer 2 Report

Thank you for the opportunity to review this manuscript. 

The Introduction need to be extended and focused more to highlight the main knowledge related to your topic.
The conclusions need to be rewriten to highlight the main findings of your systematic review. I recommend to write the future directions of the researches according with your topic.

Author Response

Dear Reviewer,

We would like to thank you for your valuable comments on the article. Below you will find our reply to your review. All changes are with a description or a comment and changes have been made to the manuscript.

The Introduction need to be extended and focused more to highlight the main knowledge related to your topic.

Thank you for your suggestion. I tried to present what the standard therapy after extraction of the third molar looks like today and where kinesiotaping is the standard nowadays.

The conclusions need to be rewritten to highlight the main findings of your systematic review. I recommend to write the future directions of the researches according with your topic.

Thank you for that remark, we changed the structure of conslusions according to your suggestion.

Reviewer 3 Report

This is an interesting systematic review, some suggestions or questions to improve the quality of the work are presented below.

In general, the review could provide much more interesting data from the selected studies and provide more information on this topic

Abstract

Line 20 What does “ (29.11.2020)” mean?

Methodology

Search strategy used for each database should be provided maybe in a table as supplementary material.

Why search was limited to the articles in the last 5 years?

If the aim of the study was: “This study aimed to 56 systematically review and synthesize controlled trials investigating the use of kinesiotaping to reduce 57 the morbidity after third molar extraction.” Why (2) cohort study; or (3) case-control study are inclusion criteria? Additionally, the methodological quality of the included studies was evaluated with the Jadad score, which the own authors indicated that is an scale for randomized controlled trials for RCT and RCCT studies. Furthermore, finally all the included studies in the review are RCT.

It would be interesting to examine the randomized clinical trials that are ongoing and are registered in databases such as clinicaltrials.gov

Results

Table 1. should be part of the results section.

I recommend including two tables, first table describing the characteristics of the studies and second describing the characteristics of the interventions.

If all the studies were RCT and were performed in human patients these two colons can be eliminated as do not provide new information.

Results presented in the table could be provided in a simpler way, not as long text. A colon describing the main outcomes could be included and the results could be presented in the next colon as improvement/increase/decrease of these outcome measures

Were the effects of the interventions of the 10 included studies evaluated with the same outcome measures? In this case, a metanalysis could be performed.

The main results of the review presented in table 1 are not commented in the results section.

Discussion

Discussion with other reviews on the same or similar topic should be provided.

Author Response

Dear Reviewer,

We would like to thank you for your valuable comments on the article. Below you will find our reply to your review. All changes are with a description or a comment and changes have been made to the manuscript

This is an interesting systematic review, some suggestions or questions to improve the quality of the work are presented below.

In general, the review could provide much more interesting data from the selected studies and provide more information on this topic

Abstract

Line 20 What does “ (29.11.2020)” mean?

We wanted to introduce the date of the search to the abstract. As suggested by another reviewer, we removed it and left the search date in materials and method section in order to avoid misunderstanding.

Methodology

Search strategy used for each database should be provided maybe in a table as supplementary material.

Thank you for your suggestion. I included the results of the query in the databases in form of .docx supplementary file. I hope it will meet your requirements. In the day of research (3rd November 2020) the study of Yurttutan and Sancak was only available on Google Scholar. However, when I checked it not, the study at the date of writing answer to the comments (18th November) also available on PubMed.

Why search was limited to the articles in the last 5 years?

Thank you for your remark. It should’ve been written “10 years”. It is caused by a misunderstanding in our team. As you can see one of the included studies came from 2014, what made no sense.

If the aim of the study was: “This study aimed to 56 systematically review and synthesize controlled trials investigating the use of kinesiotaping to reduce 57 the morbidity after third molar extraction.” Why (2) cohort study; or (3) case-control study are inclusion criteria? Additionally, the methodological quality of the included studies was evaluated with the Jadad score, which the own authors indicated that is an scale for randomized controlled trials for RCT and RCCT studies. Furthermore, finally all the included studies in the review are RCT.

This “error” came from the fact, that in my articles I always use two scales, which should be complement to each other. Usually the second one is Newcastle – Ottawa scale. This time, to be more accurate, we chose the field-specific Maastricht scale. However, it is the pure coincidence that all included studies are RCTs. As you can see in supplementary material, none other study matched our criteria than actually included 10.

It would be interesting to examine the randomized clinical trials that are ongoing and are registered in databases such as clinicaltrials.gov

One of our exclusion criteria was lack of effective statistical analysis. Such analysis cannot be done on data of ongoing studies.

Results

Table 1. should be part of the results section.

Thank you for your suggestion. We removed the table to results section.

I recommend including two tables, first table describing the characteristics of the studies and second describing the characteristics of the interventions.

Thank you for the suggestion. I agree, that the methodology of the studies should be described in more detailed way. However, after removal of one column, there is much more space to described it and leave in a form of one table, what leaves this section as more compact.

If all the studies were RCT and were performed in human patients these two colons can be eliminated as do not provide new information.

Thank you for your remark. We eliminated the colon describing the subject type/ However, we would like to leave the type of study column, as some studies are not only RCT, but also split-mouth. It is also important to underline, that all of the included studies are RCTs, what enlarge the importance of results.

Results presented in the table could be provided in a simpler way, not as long text. A colon describing the main outcomes could be included and the results could be presented in the next colon as improvement/increase/decrease of these outcome measures

Were the effects of the interventions of the 10 included studies evaluated with the same outcome measures? In this case, a metanalysis could be performed.

The outcome measures are similar, but the characteristics of interventions differ to much from each other. Some studies investigate the effect size to the different type of pharmacological therapy as control, other used drain and the some did not do any type of intervention. That is why calculating the effect size of the use of kinesiology taping is impossible.

The main results of the review presented in table 1 are not commented in the results section.

Thank you for your suggestion. As also suggested by you earlier, we replaced the table 1 to the results section.

Discussion

Discussion with other reviews on the same or similar topic should be provided.

As stands written in introduction section, “There are many RCTs suggesting a positive effect of kinesiotape application on postoperative swelling in a variety of clinical cases[12]. In dentistry, however, the use of kinesiotaping still seems to be innovative, since not many articles on kinesiotaping within the craniofacial area have been published to date, and there is no systematic review of the literature in this field at the moment of writing this paper.”

Because of the numer of changes, that we had to introduce, we did not mark changes in red in order not to make manuscript unreadable.

Round 2

Reviewer 2 Report

The authors improved the manuscript according with the recommnedations. 

Author Response

Thank you for your help in improving our manuscript.

Reviewer 3 Report

The authors have improved the quality of the manuscript as most of the suggestions have been included. However, some details should be revised:

When I stated “Search strategy used for each database should be provided maybe in a table as supplementary material.” I referred to the search strategy used in each database, not to the results obtained in each database. The supplementary material provided is presented in a disorganized way, it is difficult to follow.

Regarding the comment “Results presented in the table could be provided in a simpler way, not as long text. A colon describing the main outcomes could be included and the results could be presented in the next colon as improvement/increase/decrease of these outcome measures”

I think that the table has now improved; however, the last colon could be improved. Results could be expressed as improvement/increase/decrease of the referred outcome measures in the previous colon. Only some of the results of the outcome measures are described in the last colon.

Author Response

Dear Reviewer, 

We would like to thank you for your valuable comments on the article. Below you will find our reply to your review. All changes are with a description or a comment and changes have been made to the manuscript. 

1.When I stated “Search strategy used for each database should be provided maybe in a table as supplementary material.” I referred to the search strategy used in each database, not to the results obtained in each database. The supplementary material provided is presented in a disorganized way, it is difficult to follow.  

We added the search strategy used in each database on the Flow Diagram. Flow diagram is part of the manuscript and these precious information will be fully available to the journal’s readership at a first gaze. We also corrected some typing errors. Please find all the corrections made in red type. 

2. Regarding the comment “Results presented in the table could be provided in a simpler way, not as long text. A colon describing the main outcomes could be included and the results could be presented in the next colon as improvement/increase/decrease of these outcome measures” 

I think that the table has now improved; however, the last colon could be improved. Results could be expressed as improvement/increase/decrease of the referred outcome measures in the previous colon. Only some of the results of the outcome measures are described in the last colon. 

Thank you for comment. The last colon has been improved and facilitated according to your suggestion.